# Revisiting Point Cloud Classification with a Simple and Effective Baseline

## Abstract

Processing point cloud data is an important component of many real-world systems. As such, a wide variety of point-based approaches have been proposed, reporting steady benchmark improvements over time. We study the key ingredients of this progress and uncover two critical results. First, we find that auxiliary factors like different evaluation schemes, data augmentation strategies, and loss functions, which are independent of the model architecture, make a large difference in performance. The differences are large enough that they obscure the effect of architecture. When these factors are controlled for, PointNet++, a relatively older network, performs competitively with recent methods. Second, a very simple projection-based method, which we refer to as SimpleView, performs surprisingly well. It achieves on par or better results than sophisticated state-of-the-art methods on ModelNet40 while being half the size of PointNet++. It also outperforms state-of-the-art methods on ScanObjectNN, a real-world point cloud benchmark, and demonstrates better cross-dataset generalization.

## 1 Introduction

Processing 3D point cloud data accurately is crucial in many applications including autonomous driving (Navarro-Serment et al., 2010) and robotics (Rusu et al., 2009). In these settings, sensors like LIDAR produce unordered sets of points that correspond to object surfaces. Correctly classifying objects from this data is important for 3D scene understanding (Uy et al., 2019). While classical approaches for this problem have relied on hand-crafted features (Arras et al., 2007), recent efforts have focused on the design of deep neural networks (DNNs) to learn features directly from raw point cloud data (Qi et al., 2017a). Deep learning-based methods have proven effective in aggregating information across a set of 3D points to accurately classify objects.

The most widely adopted benchmark for comparing methods for point cloud classification has been ModelNet40 (Wu et al., 2015b). The accuracy on ModelNet40 has steadily improved over the last few years from 89.2% by PointNet (Qi et al., 2017a) to 93.6% by RSCNN (Liu et al., 2019c) (Fig. 1). This progress is commonly perceived to be a result of better designs of network architectures. However, after performing a careful analysis of recent works we find two surprising results. First, we find that auxiliary factors including differing evaluation schemes, data augmentation strategies, and loss functions affect performance to such a degree that it can be difficult to disentangle improvements due to the network architecture. Second, we find that a very simple projection-based architecture works surprisingly well, outperforming state-of-the-art point-based architectures.

In deep learning, as results improve on a benchmark, attention is generally focused on the novel architectures used to achieve those results. However, there are many factors beyond architecture design that influence performance including data augmentation and evaluation procedure. We refer to these additional factors as a method's `protocol`. A protocol defines all details orthogonal to the network architecture that can be controlled to compare differing architectures. Note that it is possible for some specific form of loss or data augmentation to be tied to a specific architecture and inapplicable to other architectures. In these cases, it would be inappropriate to treat them as part of the protocol. However, for all the methods we consider in this paper, their losses and augmentation schemes are fully compatible with each other and can be considered independently.

We do experiments to study the effect of protocol and discover that it accounts for a large difference in performance, so large as to obscure the contribution of a novel architecture. For example, the

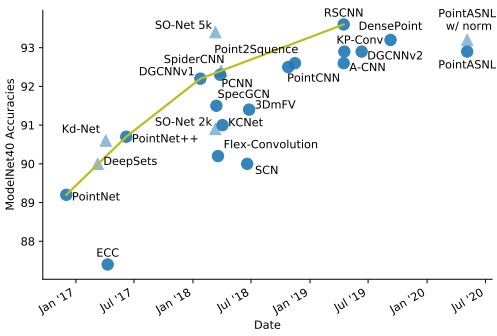 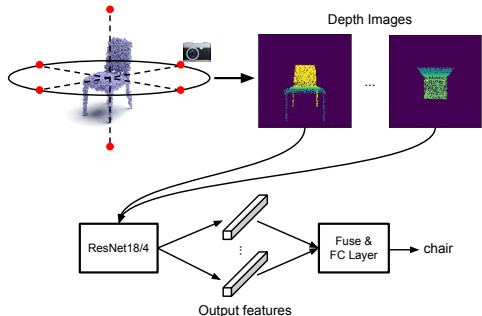

Figure 1: Performance of point-based models on ModelNet40. Those using $> 1024$ points or normals are marked with triangle. Line joins the top-performing models across times.

Figure 2: The SimpleView Architecture. The depth images are colored only for illustration. SimpleView takes in single channel depth images as input.

performance of the PointNet++ architecture (Qi et al., 2017b) jumps from $90.0 \pm 0.3$ to $93.3 \pm 0.3$, when switching from its original protocol to RSCNN's protocol (Liu et al., 2019c). We further find that the protocols that lead to the strongest performance rely on feedback from the test set, which differs from conventional evaluation setups. We re-evaluate prior architectures using the best augmentation and loss functions, while not using any feedback from the test set. We find that by taking protocol into account, the PointNet++ architecture performs competitively with more recent ones in various settings.

In addition to the surprising importance of protocol, in reviewing past approaches, another surprising discovery is that a very simple projection based baseline works very well. One needs to simply project the points to depth maps along the orthogonal views, pass them through a light-weight CNN and fuse the features. We refer to this baseline as SimpleView.

Compared to previous projection-based method (Roveri et al., 2018; Sarkar et al., 2018) for point-cloud classification, SimpleView is very simple. Prior methods have developed special modules for view selection, rendering, and feature merging, as well as use larger CNN backbones that are pretrained on ImageNet (refer to Sec. 2 for more details). In contrast, SimpleView has no such special operations, and only requires simple point projections, a much smaller CNN backbone, and no ImageNet pretraining.

The discovery of SimpleView is surprising because recent state-of-the-art results have all been achieved by point-based architectures of increasing sophistication. In recent literature, it is often assumed that point-based methods are the superior choice for point-cloud processing as they "do not introduce explicit information loss" (Guo et al., 2020). Prior work has stated that "convolution operation of these methods lacks the ability to capture nonlocally geometric features" (Yan et al., 2020), that a projection-base method "often demands a huge number of views for decent performance" (Liu et al., 2019c), and that projection-based methods often "fine-tune a pre-trained image-based architecture for accurate recognition" (Liu et al., 2019c). It is thus surprising that a projection-based method could achieve state-of-the-art results with a simple architecture, only a few views, and no pretraining.

On ModelNet40, SimpleView performs on par or better than more sophisticated state-of-the-art networks across various protocols, which includes the ones used by prior methods (Table. 3) as well as our protocol (Table. 5). At the same time, SimpleView outperforms state-of-the-art architectures on ScanObjectNN (Uy et al., 2019), a real-world dataset where point clouds are noisy (background points, occlusions, holes in objects) and are not axis-aligned. SimpleView also demonstrates better cross-dataset generalization than prior works. Furthermore, SimpleView uses less parameters than state-of-the-art networks (Table. 5).

Note that we are not proposing a new architecture or method, but simply evaluating a simple and strong projection-based baseline for point-cloud classification that is largely ignored in the literature. We do not claim any novelty in the design of SimpleView because all of its components have appeared in the literature. Our contribution is showing that such a simple baseline works surprisingly well, which is a result absent in existing literature.

It is worth noting that one might think that projection-based methods are not directly comparable with point-based methods because projection-based methods may have the full mesh as input, as opposed to just a point cloud. While this is true for existing results in the literature, it is not the case with SimpleView, whose input is the exact same point cloud given to a point-based method. In other words, SimpleView is directly comparable to a point-based method because they solve the exact same task.

In summary, our contributions are threefold:
- We show that training and evaluation factors independent of network architecture have a large impact on point-cloud classification performance. With these factors controlled for, PointNet++ performs as well as more recent architectures.
- We demonstrate how SimpleView, a very simple projection based baseline performs surprisingly well on point-cloud classification. It performs on par with or better than prior networks on ModelNet40 while using fewer parameters. It also outperforms state-of-the-art methods on real-world point-cloud classification and achieves better cross-dataset generalization.

## 2 RELATED WORK

**Point-Based Methods for Point-Cloud Analysis:** A broad class of DNNs have emerged to process 3D points directly (Simonovsky & Komodakis, 2017; Zaheer et al., 2017; Klokov & Lempitsky, 2017; Xu et al., 2018; Atzmon et al., 2018; Wang et al., 2018a; Li et al., 2018a; Groh et al., 2018; Ben-Shabat et al., 2018; Xie et al., 2018; Li et al., 2018b; Liu et al., 2019a; Thomas et al., 2019; Komarichev et al., 2019; Liu et al., 2019b; Yan et al., 2020; Su et al., 2018; Zhang et al., 2019; Liu et al., 2019a; Atzmon et al., 2018). PointNet (Qi et al., 2017a) proposed one of the first strategies, where features are updated for each point with MLP layers, and aggregated with global max pooling. However, no local comparisons are performed in PointNet, which motivates PointNet++(Qi et al., 2017b). PointNet++ breaks subsets of points into local regions that are processed first. More explicit modeling of the spatial relations between points is performed with more recent methods (Li et al., 2018b; Liu et al., 2019c; Wu et al., 2019). For example, PointConv learns functions to define continuous 3D convolutions that can be applied to arbitrary sets of points in a neighborhood (Wu et al., 2019). RSCNN uses MLPs conditioned on the spatial relationship of two points to update and aggregate features around an individual sampled point (Liu et al., 2019c). There exist many variations to these methods, but the emerging trend is an increase in sophistication.

**Projection-Based Methods for Point-Cloud Classification:** Projection-based methods for point cloud classification have been proposed in the literature. Notably, Roveri et al. (2018) learn to predict viewing angles and classify images in an end-to-end differentiable way. They use the ResNet50 model, pretrained on ImageNet as their backbone and a depth-image generation pipeline. Sarkar et al. (2018) propose a special multi-height rendering and feature merging scheme, and use a larger backbone network pretrained on ImageNet. Ahmed et al. (2019) manually define important views for each object category, create binary edge maps, and train an ensemble of PointNet++ and CNN. However, numbers in Ahmed et al. (2019) are not directly comparable to other approaches as there is a manual alignment of objects in the test set which is different from the standard ModelNet40 test set. This was confirmed with the authors. It is worth noting that even though prior work has shown sophisticated operations to be useful for achieving good results, we find that when controlling for method protocols, strong performance can be achieved with fixed orthogonal views, a smaller network, no ImageNet pretraining, and simpler rendering of points.

**Projection-Based Methods for Other Point-Cloud Analysis Tasks:** There is a rich literature for using projection-based methods on various point-cloud analysis problems like segmentation (Ladický et al., 2010; Tighe & Lazebnik, 2010; Riemenschneider et al., 2014; Qin et al., 2018; Dai & Nießner, 2018; Kalogerakis et al., 2017; Tatarchenko et al., 2018), reconstruction (Pittaluga et al., 2019) and rendering (Aliev et al., 2019). Notably, Boulch et al. (2017) use point cloud density to create scene meshes, which are then put into a mesh renderer to generate many image views at different scales. Lawin et al. (2017) render a scene point cloud from 120 views for different modalities like color, depth, and surface normal. Information from multiple modalities is then fused to generate point-wise predictions. For a detailed survey of various projection approaches on different point-cloud processing tasks, we encourage readers to check the recent survey paper by (Guo et al., 2020). In this work, SimpleView serves as a stripped-down projection-based baseline for point-cloud classification that uses a few orthogonal views and simple point projections.

Table 1: Summary of various protocols.

| Protocol | Data Augmentation | Model Selection | Loss | Ensemble | Training Points |
|---|---|---|---|---|---|
| *PointNet++* | jitter, random rotation, random scaling and trans. | final model | cross-entropy | Rotation Vote | fixed |
| *DGCNN* | random scaling and trans. | best test model | smooth-loss | No vote | fixed |
| *RSCNN* | random scaling and trans. | best test model | cross-entropy | Repeated Scaling Vote | resampled |
| *SimpleView* | random scaling and trans. | final model | smooth-loss | No vote | fixed |

**3D shape Analysis using Rendered Images and Voxels:** Many works use images rendered from object meshes for 3D shape analysis (Maturana & Scherer, 2015; Wu et al., 2015b; Yu et al., 2018; Guo et al., 2016; Shi et al., 2015; Hackel et al., 2017; Song & Xiao, 2016; 2014; Huang & You, 2016; Tchapmi et al., 2017). MVCNN exemplifies this strategy by applying a shared CNN to many rendered views and max-pooling to aggregate features (Su et al., 2015). Subsequent approaches include RotationNet which trains the network to also predict the viewpoint for each image (Kanezaki et al., 2018), GVCNN which groups features from subsets of views together before aggregating into a final prediction (Feng et al., 2018), and hypergraph methods that consider the correlation across training samples (Zhang et al., 2018; Feng et al., 2019). One notable exception is Qi et al. (2016), who use a multi-resolution variant of MVCNN, but instead of object meshes, use a voxelized version of the object for rendering. In contrast to the prior view-based methods that use object meshes with point connectivity information, and render images using basic shading and/or depth; SimpleView takes as input raw point clouds.

Another class of methods is voxel-based methods that convert points to a fixed 3D grid instead, which enables the use of 3D CNNs (Qi et al., 2016; Wu et al., 2015a; Maturana & Scherer, 2015). Given the added dimension, such methods are usually restricted to a much lower resolution to represent objects. Though some strategies such as octrees have been used to address those limitations (Wang et al., 2017), the advantages to processing 3D data directly in this manner do not yet appear to outweigh the additional overhead introduced.

## 3 METHOD OVERVIEW

### 3.1 VARIATIONS IN EXISTING PROTOCOLS

We analyze the key ingredients in the progress in point-cloud classification. Critical to our study is controlling for factors which are independent of network architecture. We refer to the factors as a method's `protocol`. A protocol used by one method can be transferred to another. For our study, we analyze a subset of the highest performing methods over the past few years. This choice was further based on availability and usability of official source-code. Specifically, we choose PointNet (Qi et al., 2017a), PointNet++ (Qi et al., 2017b), DGCNN (Wang et al., 2018b) and RSCNN (Liu et al., 2019c). Note that we also do direct comparisons to networks apart from the ones mentioned here (Table 4).

For our purposes, we do not consider any variations in input, namely the use of surface normals or more than 1024 points. Using normals or more points have been shown to improve performance in the literature. Our objective is to study factors that are not commonly perceived as a major source of performance increase. So we scope our analysis to the most widely adopted input scheme which uses 1024 points with only $x, y, z$ coordinates.

**Data Augmentation:** Various data augmentation strategies like `jittering`, `random rotation along y-axis`, `random scaling` and `random translation`. Different methods use different combinations of these augmentations. PointNet and PointNet++ use all the above augmentations. However, as objects in ModelNet40 are aligned, `random rotation along y-axis` adversely affects the performance of a model. Hence recent methods, including RSCNN and DGCNN, do not use it. They use only `random translation` and `random scaling`. Some methods including PointCNN make a distinction between whether or not `random rotation along y-axis` is used, but it is not a common practice.

**Input Points:** PointNet and PointNet++ use a fixed set of 1024 points per object to train the network. We refer to it as the `fixed points` strategy. RSCNN and PointCNN randomly sample points

Table 2: DGCNN aug. and smooth loss improve the performance of all architectures

| Data Augmentation | | Model Selection | | Loss | | Architecture | | | |
|---|---|---|---|---|---|---|---|---|---|
| PN++ | DGCNN | Final | Best Test | C.E. | Smooth | PointNet | PN++ | DGCNN | RSCNN |
| ✓ | | | ✓ | ✓ | | 89.7±0.3 | 91.0±0.3 | 90.5±0.2 | 90.4±0.3 |
| ✓ | | ✓ | | ✓ | | 89.0±0.2 | 89.8±0.2 | 90.0±0.4 | 89.4±0.1 |
| | ✓ | ✓ | | ✓ | | 89.1±0.2 | 92.1±0.1 | 91.1±0.3 | 91.1±0.3 |
| | ✓ | ✓ | | | ✓ | 89.2±0.9 | 92.7±0.1 | 91.9±0.2 | 91.7±0.3 |

during each epoch, effectively exposing the model to more than 1024 points per object during the training process. We refer to this as the `resampled points` strategy.

**Loss Function:** `cross-entropy` (CE) is used by most of the methods. However, DGCNN uses `smooth-loss`, where the ground-truth labels are smoothed out before calculating cross-entropy. We observe that `smooth-loss` improves the performance of all network architectures.

**Selecting Model for Testing:** PointNet and PointNet++ use the final converged model to evaluate on the test set. Since the number of epochs is a hyper-parameter that depends on factors like data, model, optimizer, and loss, in our experiments, we create a validation set from the training set to tune the number of epochs. We then retrain the model with the complete training set to the tuned number of epochs. We refer to this strategy as `final model selection`. We find that some methods including DGCNN and RSCNN evaluate the model on the test set after every epoch and use the best test performance as the final performance. We refer to this strategy as `best test model selection`.

**Ensemble Scheme:** Some methods use an ensemble to further improve the performance. PointNet and PointNet++ apply the final network to multiple rotated and shuffled versions of the point cloud, and average the predictions to make the final prediction. We refer to this strategy as `Rotation Vote`. The shuffling operation induces randomness in prediction for PointNet++ and RSCNN as they are not strictly invariant to the order of the points (Sec. 3.3 in Qi et al. (2017b)). Hence, while evaluating `Rotation Vote`, we do the inference 10 times per run for PointNet++ and RSCNN, and report mean and standard deviation. SimpleView and PointNet are invariant to the order of the points and hence are not affected by shuffling. Some methods, including RSCNN and DensePoint, create multiple randomly scaled and randomly sampled versions of a test object. They then evaluate the final network on these multiple versions of the object and average the prediction. Since the scaling is random, it makes the test set performance random as well. RSCNN and DensePoint repeat this procedure 300 times on the test set and report the best accuracy. We refer to it as `Repeated Scaling Vote`. DGCNN does not use any ensemble.

Table 1 summarizes the *PointNet++*, *DGCNN* and *RSCNN* protocols. Besides these three protocols, we also include variants of these protocols in table 3 and table 4, such as *PointNet++ no Vote* (i.e. *PointNet++* but without the `Rotation Vote`), *DGCNN CE* (i.e. *DGCNN* but with `CE loss` intead of `smooth loss`), *DGCNN CE Final* (i.e. *DGCNN CE* but with `final model selection` instead of `best test model selection`) and *RSCNN no Vote* (i.e. *RSCNN* but without the `Rotation Vote`). These protocols represent prototypical settings and have been used with slight modifications in many other prior works.

For example, DeepSets (Zaheer et al., 2017) used the *PointNet++ no Vote* protocol without jittering and translation; SO-Net (Li et al., 2018a) used the *DGCNN CE* protocol with jittering and random scaling instead of random scaling and translation; 3DmFV (Ben-Shabat et al., 2018) used the *DGCNN CE* protocol with additional jittering; PCNN (Atzmon et al., 2018) used the `DGCNN CE Final` protcol[1]; PointCNN (Li et al., 2018b) used the *DGCNN CE* protocol with randomly sampled points and small (10°) rotation augmentation; DensePoint (Liu et al., 2019b) used the *RSCNN* protocol; PointASL (Yan et al., 2020) used the *DGCNN CE* protocol but with additional point jittering augmentation and voting.

**Our Protocol:** Based on our findings, we define our *SimpleView* protocol, which uses the best augmentation and loss functions while not using any information from the test set. Table 2 shows that

---

[1] It is unclear from code if the best test or final model selection is used. We assume final model selection to err on the side of caution.

Table 3: Performance of various architectures on ModelNet40. Protocol affects performance by a large amount. SimpleView performs on par or better than prior architectures across protocols.

| Protocol → | PointNet++ | | RSCNN | | DGCNN | |
|---|---|---|---|---|---|---|
| Architecture ↓ | no Vote | Vote | no Vote | Vote | CE | Smooth |
| PointNet | 89.0 ± 0.2 | 89.1 ± 0.2 | 90.0 ± 0.3 | 90.1 ± 0.2 | 90.1 ± 0.2 | 90.5 ± 0.1 |
| PointNet++ | 89.8 ± 0.2 | 90.0 ± 0.3 | 92.7 ± 0.1 | **93.3** ± 0.3 | 92.6 ± 0.2 | 93.1 ± 0.2 |
| DGCNN | 90.0 ± 0.4 | 90.5 ± 0.4 | 92.2 ± 0.1 | 92.8 ± 0.5 | 91.9 ± 0.2 | 92.7 ± 0.1 |
| RSCNN | 89.4 ± 0.1 | 90.2 ± 0.2 | 92.1 ± 0.1 | 92.5 ± 0.2 | 91.9 ± 0.2 | 92.5 ± 0.1 |
| SimpleView | **90.7** ± 0.3 | **91.0** ± 0.2 | **92.9** ± 0.2 | 93.2 ± 0.1 | **93.1** ± 0.1 | **93.6** ± 0.3 |

Table 4: Performance of various architectures on ModelNet40. Includes prior works not in Table 3. * indicates small differences in protocol as identified in Sec. 3.1

| Architecture | # Points | Closest Protocol | Acc. | PN++ Acc. | SimpleView Acc. |
|---|---|---|---|---|---|
| DeepSets (Zaheer et al.) | 5000 | PointNet++ no Vote* | 90.0 ± 0.3 | 89.8 ± 0.2 | 90.7 ± 0.3 |
| SO-Net (Li et al.) | 2048 | DGCNN CE* | 90.9 | 92.6 ± 0.2 | 93.1 ± 0.1 |
| 3DmFV (Ben-Shabat et al.) | 1024 | DGCNN CE* | 91.4 | 92.6 ± 0.2 | 93.1 ± 0.1 |
| PCNN (Atzmon et al.) | 1024 | DCNN CE Final* | 92.3 | 92.1 ± 0.1 | 92.5 ± 0.3 |
| PointCNN (Li et al.) | 1024 | DGCNN CE* | 92.5 | 92.6 ± 0.2 | 93.1 ± 0.1 |
| DensePoint (Liu et al.) | 1024 | RSCNN no Vote | 92.8 | 92.7 ± 0.1 | 92.9 ± 0.2 |
| RSCNN-Multi (Liu et al.) | 1024 | RSCNN no Vote | 92.9 | 92.7 ± 0.1 | 92.9 ± 0.2 |
| PointANSL (Yan et al.) | 1024 | DGCNN CE* | 92.9 | 92.6 ± 0.2 | 93.1 ± 0.1 |

DGCNN's augmentation (i.e `random translation and scaling`) and `smooth-loss` improve performance of all prior networks, so we use them in the *SimpleView* protocol. Further, similar to PointNet, PointNet++ and DGCNN, we use the fixed dataset of 1024 points instead of re-sampling different points at each epoch. Re-sampling points for each epoch effectively increases the training dataset of points, making numbers incomparable to methods using a fixed dataset. We avoid any feedback from the test set and use the `final model selection`, where we first tune the number of epochs on the validation set then retrain the model on the entire train set. Lastly, similar to DGCNN, we do not use ensemble as it is more standard in Machine Learning to compare models without ensemble.

## 3.2 SIMPLEVIEW

Given a set of points SimpleView, projects them onto the six orthogonal planes to create sparse depth images. It then extracts features from the depth images using a CNN and fuses, which is then used to classify the point-cloud as shown in Fig. 1.

**Generating Depth Images from Point Cloud:** Let $(x, y, z)$ be the coordinates of a point in the point cloud with respect to the camera. We apply perspective projection to get the 2D coordinate $(\tilde{x} = x/z, \tilde{y} = y/z)$ of $\mathbf{p}$ at depth $z$. We also do ablations with orthographic projection and found perspective projection to work better (Table. 7). Since coordinates on image plane have to be discrete, we use $(\lceil \tilde{x} \rceil, \lceil \tilde{y} \rceil)$ to be the final coordinate of $\mathbf{p}$ on the image plane. Multiple points may be projected to the same discrete location on the image plane. To produce depth value at an image location, we do ablations on two choices, one the minimum depth of all points, and other weighted average depth with more weight ($\frac{1}{z}$) given to closer points (Table. 7). Empirically, we find both perform similar with the later performing slightly better. This could be because of reduction in noise due to the averaging of nearby pixels on the surface. The depth images are of resolution 128 X 128.

**SimpleView Architecture:** To make the number of parameters comparable to point-based methods, we use ResNet18 with one-fourth filters (ResNet18/4) as the backbone. For fusing features, we do ablation with two choices, pooling and concatenation. Empirically, we find concatenation to work better than pooling them (Table. 7). This could be because pooling features throws away the view information like which views are adjacent to one another. One concern could be that concatenation could make features sensitive to viewpoint, and hence the network could fail on rotated objects.

Table 5: Performance of various architectures on ModelNet40 when using the best data-augmentation and loss function; and not using any feedback from test set. SimpleView outperforms prior architectures, while having fewest parameters and comparable inference time.

| Architecture ↓ | Acc. | Class Acc. | Para. (M) | Time (ms) |
|---|---|---|---|---|
| PointNet | 89.2 ± 0.9 | 85.1 ± 0.6 | 3.5 | **3.0** |
| PointNet++ | 92.7 ± 0.1 | 90.0 ± 0.3 | 1.7 | 20.7 |
| DGCNN | 92.3 ± 0.3 | 89.1 ± 0.3 | 1.8 | 7.3 |
| RSCNN | 91.7 ± 0.3 | 88.5 ± 0.4 | 1.3 | 4.3 |
| SimpleView | **93.0** ± 0.4 | **90.5** ± 0.8 | **0.8** | 5.0 |

Table 6: Performance of various architectures on ScanObjectNN, and cross-dataset generalization. SimpleView achieves state-of-the-art results and shows better cross dataset generalization. Numbers for prior works are from (Uy et al., 2019).

| Architecture ↓ | TR: SONN TE: SONN | TR: MN40 TE: SONN | TR: SONN TE: MN40 |
|---|---|---|---|
| 3DmFV (Shabat et al.) | 63.0 | 24.9 | 51.5 |
| PointNet (Qi et al.) | 68.2 | 31.1 | 50.9 |
| SpiderCNN (Xu et al.) | 73.7 | 30.9 | 46.6 |
| PointNet++ (Qi et al.) | 77.9 | 32.0 | 47.4 |
| DGCNN (Wang et al.) | 78.1 | 36.8 | 54.7 |
| PointCNN (Li et al.) | 78.5 | 24.6 | 49.2 |
| SimpleView | **79.5±0.5** | **40.5±1.4** | **57.9±2.1** |

Table 7: Ablation of various choices for SimpleView on ModelNet40. The performance is evaluatated on the validation set.

| | Number of Views | | | Image Projection | | Feature Fusion | | Image Depth | |
|---|---|---|---|---|---|---|---|---|---|
| | 1 | 3 | 6 | Orthographic | Perspective | Pool | Concat | Minimum | Weighted Avg. |
| Accuracy | 90.7 ± 0.1 | 92.1 ± 0.2 | 92.9 ± 0.3 | 92.7 ± 0.3 | 92.9 ± 0.3 | 91.8 ± 0.3 | 92.9 ± 0.3 | 92.8 ± 0.4 | 92.9 ± 0.3 |

However, empirically, we observe that this issue is largely mitigated by rotation augmentation and SimpleView is able to achieve state-of-the-art performance on ScanObjectNN where objects are rotated. The point-clouds are scaled to be in $[1, -1]^3$, we keep the cameras at a distance of $1.4$ units from the center with $90°$ fov. We also do ablations with different number of views, comparing only front views, three orthogonal views and six orthogonal views. We find that using all six views performs the best (Table 7). We do not use ImageNet pretraining, thus making the comparison with point-based methods strictly fair, without any additional data.

# 4 EXPERIMENTS

**ModelNet40:** ModelNet40 is a the most widely adopted benchmark for point-cloud classification. It contains objects from 40 common categories. There are 9840 objects in the training set and 2468 in the test set. Objects are aligned to a common up and front direction.

**ScanObjectNN:** ScanObjectNN is a recent real-world point cloud classification dataset. It consists of 15 classes, 11 of which are also in ModelNet40. There are a total of 15k objects in the dataset. Unlike ModelNet40, the objects in ScanObjectNN are obtained from real-world 3D scans. Hence, point clouds are noisy (occlusions, background points) and have geometric distortions such as holes. Also, unlike ModelNet40, the objects are not axis-aligned.

## 4.1 EXPERIMENTS ON MODELNET40

**Implementation Details:** We use PyTorch (Paszke et al., 2019) to implement all models and protocols while reusing the official code wherever possible. We use the official version of DGCNN and RSCNN. We confirm with the authors that the code for RSCNN-Multi, another version of RSCNN, is yet to be released. Hence we use the reported numbers of RSCNN-Multi in Table 4. PointNet and PointNet++ are officially released in TensorFlow (Abadi et al., 2015). For PointNet, we adapt our code from PointNet.pytorch (Xia, accessed June, 2020) as recommended in the official repository. For PointNet++, we adapt the model code from Pointnet2_PyTorch (Wijmans, accessed June, 2020). We further make sure that the third party PyTorch code closely matches the official TensorFlow code.

We use Adam (Kingma & Ba, 2014) with an initial learning rate of 1e-3 and a decay-on-plateau learning rate scheduler. The batch size and weight decay for each model are kept the same as the official version in Table 3. We use a batch size of 18 and no weight decay for SimpleView. To give the prior models the best chance on our protocol (Table 5), we additionally tune their hyper-parameters on the validation set. We find that the official hyper-parameters already perform close to

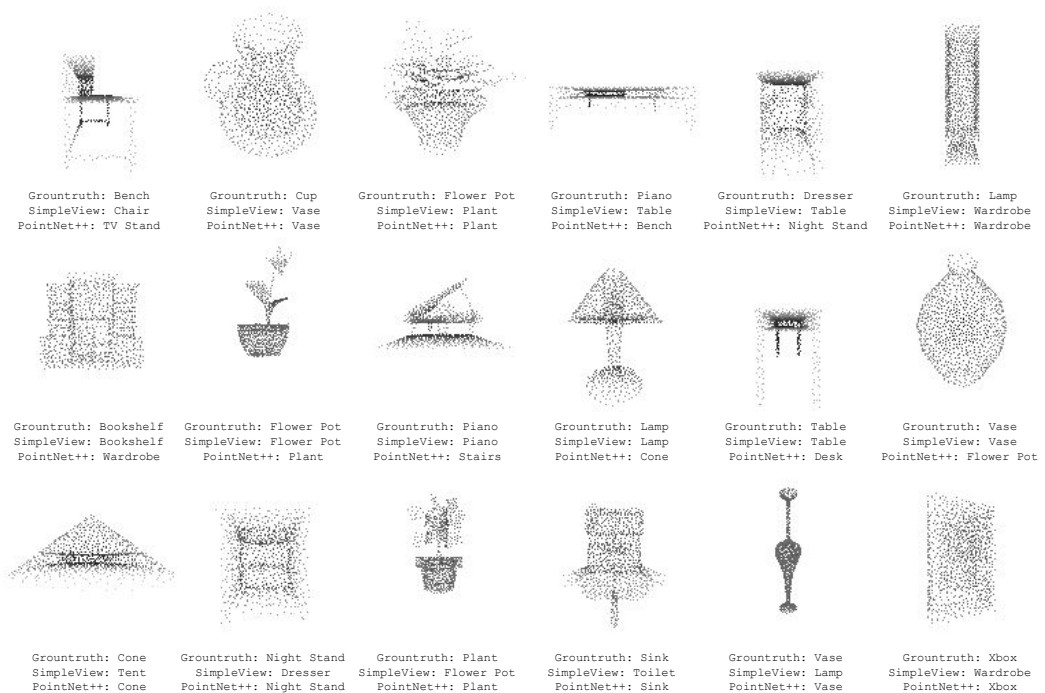

Figure 3: Failure Cases for SimpleView and PointNet++. The first row shows cases where both SimpleView and PointNet++ fail; the second row shows cases where SimpleView succeeds but PointNet++ fails; the third row shows cases where SimpleView fails but PointNet++ succeeds.

optimal. We train each model for 1000 epochs. Since there are small variations in final performance across different runs, we do 4 runs and report the mean and standard deviation.

**Performance under various Prior Protocols:** Table 3 shows the performance different architectures under various protocols. The mean performance of PointNet++ improves from 89.8% to 93.3% when we switch from the *PointNet++ no Vote* to the *RSCNN Vote* protocol. Similarly the performance of SimpleView improves from 90.7% to 93.6% when we switch from *PointNet++ no Vote* to *DGCNN Smooth*. Since there is variance in performance across runs, we refrain from making any claims about absolute ordering between prior works. However, we do observe that in terms of mean performance, SimpleView performs on par or better than other methods under all protocols. Note that in *RSCNN Vote*, voting on the test set is done 300 times with reshuffled and randomly augmented points, from which the highest accuracy is selected. Hence models that have the largest variance in prediction, i.e. PointNet++ and RSCNN gain the most from it, as they are not strictly invariant to the order of points (Sec. 3.3 in Qi et al. (2017b)).

**Performance under the *SimpleView* Protocol:** Table 5 shows that SimpleView outperforms prior architectures on our controlled protocol in terms of mean performance. SimpleView has the fewest number of parameters and a competitive inference speed. Inference speed is measured on an NVIDIA 2080Ti averaged across 100 runs.

Fig. 3 show examples where both SimpleView and PointNet++ fail, as well as examples where one of them fails and the other succeeds. Qualitatively, we find that the failure modes of SimpleView and PointNet++ are similar. We also find that a major failure mode in both SimpleView and PointNet++ is the confusion between the 'flower_pot' and 'plant' category (see Sec. A Fig. I and Fig. II). This could be because of the lack of color information.

**Comparison with More Methods:** In Table 4, we do one-on-one comparison between SimpleView and recent state-of-the-art methods, other than PointNet, PointNet++, RSCNN and DGCNN. We identify the closest protocol to the one used in the paper from the ones we evaluate. Table 4 shows the competitiveness of PointNet++ and SimpleView with other recent state-of-the-art methods.

## 4.2 Experiments on ScanObjectNN

**Implementation Details:** ScanObjectNN's official repository trains and evaluates the state-of-the-art models under the same protocol. We implement SimpleView in TensorFlow and use the official ScanObjectNN protocol for fairness. This protocol is different from the *SimpleView* protocol as it normalizes the point clouds and randomly samples points. We optimize our model with Adam. We use a batch size of 20 and no weight decay to train SimpleView for 300 epochs with an initial learning rate 0.001, and use the final model for testing. We use standard image-based cropping and scaling augmentation to prevent over-fitting. The hyper-parameter for cropping and scaling is found on a validation set made from ScanObjecNN's train set. We conduct 4 runs for SimpleView. ScanObjectNN does not use a fixed set of points during test time. It instead randomly samples points from the point cloud, which adds randomness to test set performance. Hence, we evaluate each run 10 times. We report the final performance as the mean and variance of the 40 evaluations (4 runs × 10 evaluations per run).

**Performance on ScanObjectNN:** As shown in Table 6, SimpleView outperforms prior networks on ScanObjectNN. This shows the SimpleView is effective in real world settings, with noisy and mis-aligned point clouds. We also perform transfer experiments to test generalizability of SimpleView. We train on ScanObjectNN and test on ModelNet40 and vice versa. Table 6 shows that SimpleView transfers across datasets better than prior methods.

## 5 Discussion

In this work, we demonstrate how auxiliary factors orthogonal to the network architecture have a large effect on performance for point-cloud classification. When controlling for these factors, we find that a relatively older method, PointNet++ (Qi et al., 2017b), performs competitively with more recent ones. Furthermore, we show that a simple baseline performs on par or better than state-of-the-art architectures.

Our results show that for future progress we should control for protocols while comparing network architectures. Our code base could serve as a useful resource for developing new models and comparing them with prior works. Our results show that the existing evidence for point-based methods is not as strong when auxiliary factors are properly controlled for, and that SimpleView is a strong baseline. But our results are not meant to discourage future research on point-based methods. It is still entirely possible that point-based methods come out ahead with additional innovations. We believe it is beneficial to explore competing approaches, including the ones that are underperforming at a particular time, as long as the results are compared in a controlled manner.

Our analysis in this work was limited to point cloud classification, which is an important problem in 3D scene understanding and forms a critical part of object detection and retrieval systems. An exciting future direction would be to expand this analysis to other problems that involve point cloud data such as scene and part segmentation.

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

# A  APPENDIX

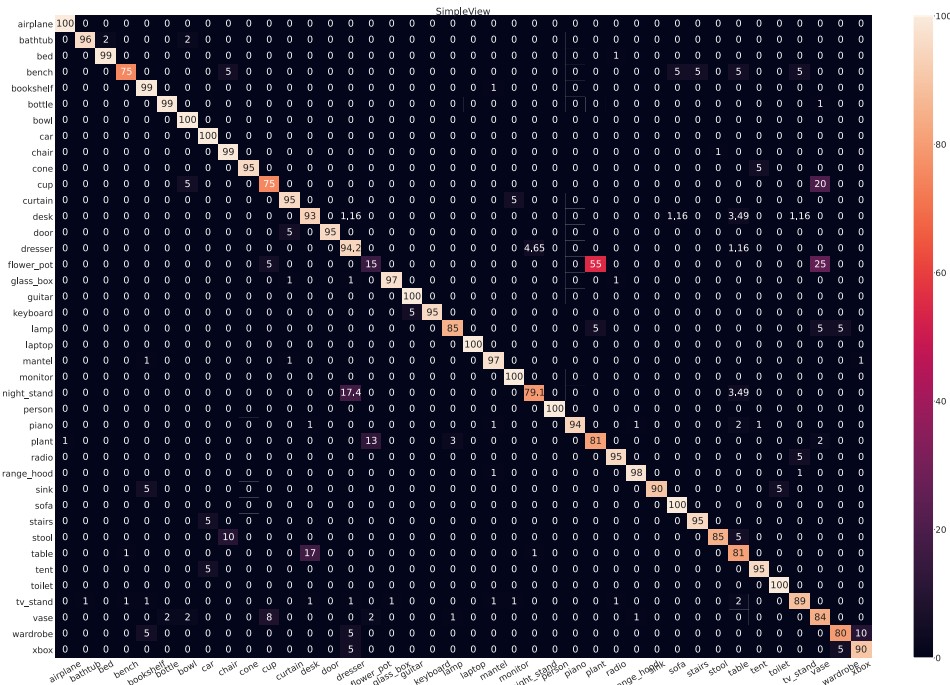

Figure I: Confusion matrix for SimpleView when trained under our protocol

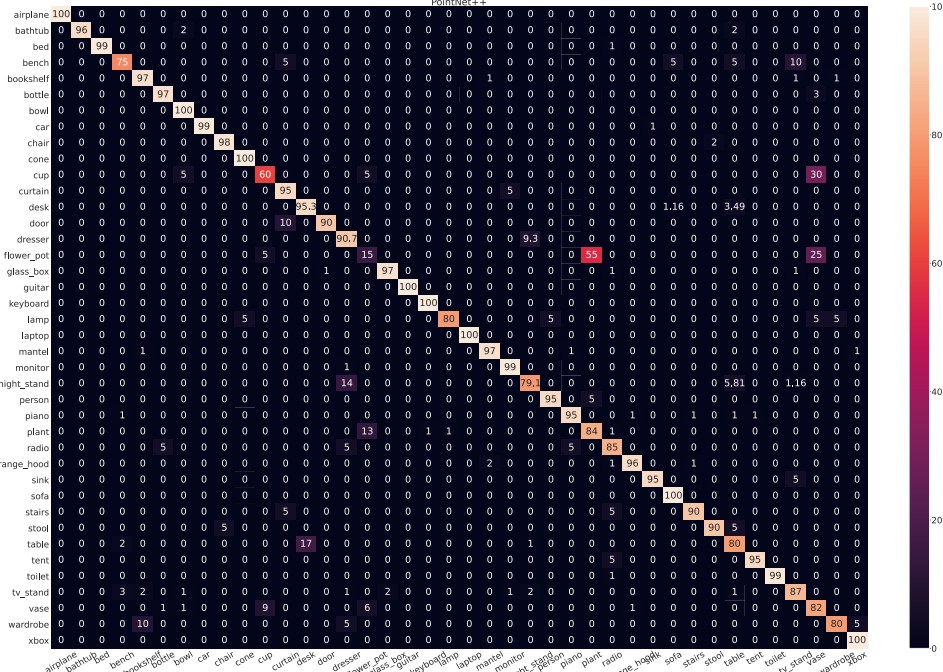

Figure II: Confusion matrix for PointNet++ when trained under our protocol

Table I: Performance of various architectures on ModelNet40 when using different amount of training data.

| Per. of Training Data | RSCNN | DGCNN | PointNet | PointNet++ | SimpleView |
|---|---|---|---|---|---|
| 25 % | 88.2 ± 0.4 | 89.1 ± 0.2 | 86.3 ± 0.4 | 89.6 ± 0.4 | 89.7 ± 0.3 |
| 50 % | 90.4 ± 0.4 | 91.0 ± 0.3 | 88.2 ± 0.3 | 91.5 ± 0.2 | 92.1 ± 0.3 |
| 100 % | 91.7 ± 0.3 | 92.3 ± 0.3 | 89.2 ± 0.9 | 92.7 ± 0.3 | 93.0 ± 0.4 |

