# OpenReview forum: "Revisiting Point Cloud Classification with a Simple and Effective Baseline"
_ICLR.cc/2021/Conference — Reject_

### Official Review · AnonReviewer4 · 2020-10-27
**An interesting and nice paper on analysing various modern point cloud processing models**

**Rating:** 7
**Confidence:** 4

**Review:**

Summary:
In this paper, the author(s) do a careful analysis on the classification performance of various modern point cloud processing networks and show empirically that with evaluation protocol set the same for different models, PointNet++, which is a relatively old model, has similar or better performance than newly proposed methods. The author(s) also show a simple projection based baseline SimpleView that can work surprisingly well on point cloud classification task. They evaluate methods on ModelNet40 and ScanObjectNN datasets.
-----------------
Pros:
1. The paper is well written and easy to follow.
2. The author(s) conduct detailed break-downs on the evaluation protocols used by various modern point processing models, which I think is valuable to research on point cloud data. It is also surprising that factors beyond architecture design can make such a difference in the evaluation. I think following the same evaluation protocol can be very helpful for subsequent research. This also indicates evaluating point processing models solely on ModelNet40 might not be a good practice since it poses bias to some augmentation/architecture, which might be false if models are applied in real-world data. These messages are worth to be known by the community.
3. The author(s) also show that a surprisingly simple baseline that makes use of different view projections can work quite well on point cloud object detection.
-----------------
Cons:
1. It would be nice if the author(s) can include some failure mode analysis on different models. Since the SimpleView does not operate directly on point clouds, it might have distinctive failure modes compared with other models. It might be helpful for designing point cloud processing models in the future.
2. It is unsure how such projection-based methods, like the MVCNN and SimpleView, handle larger scenes (e.g. with covered objects or a scene scan in autonomous driving scenarios) and per-point tasks such as part segmentations.
-----------------
Misc:
1. Will you release source code?
2. The paper format seems not complying with ICLR format, especially the font?
3. The overall figure and the projected images in figure 2 do not match？
4. Page 6 last line, choice -> choices
-----------------
Post-rebuttal review

I carefully read through the rebuttal and other reviews and I would like to keep my original rating. The author(s) addressed my concerns and I think it is a good paper for the community.

---

> ### Author Response · Authors · 2020-11-22
> **Authors' Response to Reviewer4**
>
> Thank you for your feedback and suggestions. It is encouraging that you found our work valuable for research on point cloud data; our paper well written and easy to follow; and our evaluation detailed. We address your concerns below:
>
> **“It would be nice if the author(s) can include some failure mode analysis on different models. Since the SimpleView does not operate directly on point clouds, it might have distinctive failure modes compared with other models. It might be helpful for designing point cloud processing models in the future.”**
>
> Thanks for the suggestion. We now add qualitative analysis in the paper. We show examples of examples where both SimpleView and PointNet++ fail, as well as examples where one of them fails and the other succeeds. Qualitatively, we find that the failure modes of SimpleView and PointNet++ are similar. We also find that a major failure mode in both SimpleView and PointNet++ is the confusion between the ‘flower_pot’ and ‘plant’ category, which could be because of lack of color information.
>
> **“It is unsure how such projection-based methods, like the MVCNN and SimpleView, to handle larger scenes (e.g. with covered objects or a scene scan in autonomous driving scenarios) and per-point tasks such as part segmentations.”**
>
> Actually, prior works have used projection-based methods for segmentation in large scenes, which suggests that projection-based methods could handle large scenes and per-point tasks. Although, these methods have been relatively more involved than SimpleView. Some notable ones are [1] and [2], which we also referred to in our paper. In [1], the point cloud density is used to create scene meshes, which are then rendered at different scales using a mesh renderer. In [2], a scene point cloud is rederered for different modalities like color, depth and surface normal. Information from multiple modalities is then fused to generate the point-wise predictions.
>
> [1] Alexandre Boulch, Bertrand Le Saux, and Nicolas Audebert. Unstructured point cloud semantic labeling using deep segmentation networks. 3DOR (2017).
>
> [2] Felix J¨aremo Lawin, Martin Danelljan, Patrik Tosteberg, Goutam Bhat, Fahad Shahbaz Khan, and Michael Felsberg. Deep projective 3d semantic segmentation. CAIP (2017).
>
> **“Will you release source code?”**
>
> Yes, we will release the code.
>
> **“``````The paper format seems not complying with ICLR format, especially the font?“**
>
> Thanks for pointing it out. We corrected it in the revised version.
>
> **“The overall figure and the projected images in figure 2 do not match?“**
>
> Thanks for pointing it out. We corrected it in the revised version.
>
> **“Page 6 last line, choice -> choices“**
>
> Thanks for pointing it out. We corrected it in the revised version.
>
> We would very much appreciate it if you could  let us know if any concerns still remain. Thanks again for your suggestions and help in improving our submission.

---

### Official Review · AnonReviewer1 · 2020-10-28

**Rating:** 7
**Confidence:** 4

**Review:**

This paper studies the factors that are related to point cloud classification but independent of model architecture. Then a light-weight projection-based model is proposed. Substantial experiments are conducted to show how the auxiliary factors affect the evaluation results and the proposed method can perform at similar level compared with state-of-the-art methods.

On the whole, I think this is a good paper as it addresses one of the most important problem of current research in point cloud classification.  We need to distinguish what actually brings the performance improvement: is it the evaluation scheme or the new model architecture itself. Extensive experiments show that 1) different methods indeed use different protocols and 2) data augmentation, loss function, voting play important role in the model performance. It's valuable to know what kind of data augmentation and loss function can improve performance generally in controlled setting. The paper is well written, rigorous and easy to follow. I think this paper would be a good contribution to the community.

Some questions I have:
(1)What's the reason that performances of architectures in Table3 under RSCNN/DGCNN outperforms the performance in Table4? Is it because RSCNN/DGCNN use feedback from test set?
(2)For performance in Table 6,  do you train the the state-of-the-art method with the same protocol for SimpleView?
(3)Have you tried what the performance would be given different amount of data for different architecture under same protocol?
(4)For other more sophisticated operations in related projection-based methods, have you tried to see if they are useful under same protocol?

--------
After discussion:

After reading the author's response as well as the opinions from other reviewers, I will stick to my original rating. The authors resolve most of my questions and concerns and I look forward to a revised version of the paper.

---

> ### Author Response · Authors · 2020-11-22
> **Authors' Response to Reviewer1**
>
> Thank you for your feedback and suggestions. It is encouraging that you found our work addressing one of the most important problems of current research in point cloud classification; our paper well written, rigorous and easy to follow; and our experiments extensive. We address your concerns below:
>
> **“What's the reason that performances of architectures in Table3 under RSCNN/DGCNN outperforms the performance in Table4? Is it because RSCNN/DGCNN use feedback from test set?”**
>
> Yes, it is because RSCNN/DGCNN use feedback from the test set.
>
> **“For performance in Table 6, do you train the the state-of-the-art method with the same protocol for SimpleView”**
>
> Yes, we train the state-of-the-art methods with the same protocol.
>
> **“Have you tried what the performance would be given different amount of data for different architecture under same protocol?”**
>
> Thanks for the suggestion. We now train models with different amounts of training data on our protocol. We find that all conclusions remain the same. Specifically, for 25% of the training data RSCNN gets 88.2 $\pm$ 0.4, DGCNN gets 89.1 $\pm$ 0.2, PointNet gets 86.3 $\pm$ 0.4, PointNet++ gets 89.6 $\pm$ 0.4 and SimpleView gets 89.7 $\pm$ 0.3 on the test set. Also for 50% of the training data RSCNN gets 90.4 $\pm$0.4, DGCNN gets 91.0 $\pm$ 0.3, PointNet gets 88.2 $\pm$ 0.3, PointNet++ gets 91.5 $\pm$ 0.2 and SimpleView gets 92.1 $\pm$ 0.3 on the test set. We added this information in table I in the Appendix (see the uploaded revised paper).
>
> **“For other more sophisticated operations in related projection-based methods, have you tried to see if they are useful under same protocol?”**
>
> No, we did not try adding any sophisticated operations to SimpleView. We consider SimpleView as a simple and strong baseline but not a final solution to the problem. Exploring directions to extend SimpleView would be an exciting direction for future research.
>
> We would very much appreciate it if you could  let us know if any concerns still remain. Thanks again for your suggestions and help in improving our submission.

---

### Official Review · AnonReviewer3 · 2020-10-28
**An application paper with no scientific novelty, but might bring a large impact to the community**

**Rating:** 7
**Confidence:** 4

**Review:**

Summary:

This paper systematically analyzes performance of point cloud classification methods with strict control over the setup and hyperparameters including data augmentation, loss, input representation, model epoch selection. Through extensive experiments, this paper reveals that there seem no real improvements due to the proposed model architecture in the last two years and all performance improvements come from the training/testing setup and hyperparameter tuning.

Paper Strengths:
1. This paper provides important findings of the source of performance improvements in the domain of point cloud classification. If the findings are correct, it will imply that many efforts prior work has been put in terms of innovating new network architecture are not very effective as we expected, sometimes might be less useful than using a better training/testing protocol.
2. The presentation (language and logic) of the paper is clear (though part of the reason is that there is no involving technical description ), I really enjoy reading this paper, and I appreciate the authors for their efforts.
3. The provided experiments are really extensive, with 7 tables and 2 datasets, which makes the findings quite convincing.

Paper Weaknesses:
1. As this paper has written, there is no scientific novelty in the paper. In my mind, It is totally okay to not propose any new method/architecture but it would be nice to see a solution to the revealed problem. Specifically, this paper has revealed an important problem that prior work in innovating network architecture does not seem to clearly bring improvements. However, there is no clear solution/direction proposed in the paper to this problem. What should we do as a community next? Should researchers in point cloud classification stop innovating these useless and sophisticated network architectures? Should we just limit our research by exploring different types of data augmentation? I would guess the answer to these questions is no. But then, what should we do? It would be nice if the paper can bring further insight into this direction
2. The paper’s findings/claims are mostly based on experiments, so it would be nice if the paper can release its code to be used by the community and validate if it is really true that the efforts put in neural network architecture are not useful in terms of performance in point cloud classification. Without careful validation by the community, the strong claim made in this paper could be misleading and discouraging further research for innovating new approaches in this domain
3. Although many quantitative experiments are provided, it would be nice if this paper can add more qualitative analysis to help readers understand where the improvements are.
4. I believe there are 100+ methods proposed in the recent 3 years at many conferences/journals for point cloud processing and classification, but only 4 are evaluated in this paper. It would be nice if the authors can add more representative baselines to validate the claims in order to double confirm if it is true that there is no performance gain from the architecture side. It might be possible that the claim is true within the compared 4 baselines and there might be other methods which really improves performance solely because they have a better network

Detailed Comments
1. It might be good to combine the third contribution in the introduction with the second one?
2. In the section of “our protocol”, this paper uses the fixed set of 1024 points instead of re-sampling different points at each epoch. Why? As the paper described, using the re-sampling point strategy can effectively increase the training dataset of points, which is good. It would be nice to justify why not using such a good strategy. Also, this strategy is fair unlike the one using feedback from the test set which is kind of cheating.
3. In the section of “generating depth images from point cloud”, this paper uses two strategies, one using the minimum depth of all points which makes sense to me. But the latter one which uses a weighted average of depth does not seem to be reasonable as it is essentially creating imaginary non-existing points which correspond to the averaged depth

Justification:

My decision is made mainly because I feel the findings in this paper are important and need to be shared with the community. Sometimes, people care too much about novelty and try to design more and more sophisticated networks. If it turns out these innovative networks do not really bring performance gain, we need to re-think what we should do. However, there are spaces to be improved in this paper as I pointed out above. It would be nice to see some revision in the final version if the paper is accepted, including potential solutions/directions about where we should go and more analysis (qualitative and baselines) to support the claim.

Post-rebuttal review

After reading the authors' responses and other reviews, I would like to stick to my original rating to accept this paper. Though there are minor problems that I still have concerns (e.g., why not using re-sampling for all methods which is a better strategy than using fixed points), I am satisfied with the responses to my primary concerns about the paper.

---

> ### Author Response · Authors · 2020-11-22
> **Authors' Response to Reviewer3 (Part 1/2)**
>
> Thank you for your feedback and suggestions. It is encouraging that you found our work impactful and important; our paper very clear and well presented; and our experiments extensive and convincing. We address your concerns below:
>
> **“there is no clear solution/direction proposed in the paper to this problem. What should we do as a community next? Should researchers in point cloud classification stop innovating these useless and sophisticated network architectures? Should we just limit our research by exploring different types of data augmentation? I would guess the answer to these questions is no. But then, what should we do? It would be nice if the paper can bring further insight into this direction”**
>
> Thanks for the suggestion. We added a discussion in the revised version (under Sec. 5) describing what we believe could be a future direction for point cloud classification.
>
> Specifically, our results show that for future progress we should control for protocols while comparing network architectures. Our code base could serve as a useful resource for developing new models and comparing them with prior works. Our results show that the existing evidence for point-based methods is not as strong when auxiliary factors are properly controlled for, and that SimpleView is a strong baseline. But our results are not meant to discourage future research on point-based methods. It is still entirely possible that point-based methods come out ahead with additional innovations. We believe it is beneficial to explore competing approaches, including the ones that are underperforming at a particular time,  as long as the results are compared in a controlled manner.
>
> **Release the code**
>
> Yes we will open-source our code.
>
> **“Although many quantitative experiments are provided, it would be nice if this paper can add more qualitative analysis to help readers understand where the improvements are.”**
>
> Thanks for the suggestion. We have now added qualitative analysis in the paper (a revised version has been uploaded). We show examples where both SimpleView and PointNet++ fail, as well as examples where one of them fails and the other succeeds. Qualitatively, we find that the failure modes of SimpleView and PointNet++ are similar. We also find that a major failure mode in both SimpleView and PointNet++ is the confusion between the ‘flower_pot’ and ‘plant’ category, which can be due to lack of color information.
>
> **“It would be nice if the authors can add more representative baselines to validate the claims in order to double confirm if it is true that there is no performance gain from the architecture side. It might be possible that the claim is true within the compared 4 baselines and there might be other methods which really improves performance solely because they have a better network”**
>
> Thanks for the suggestion. In the revised version, we have expanded Table 7 to add comparisons with more methods. Most of the models are officially implemented in Tensorflow or Caffe while our codebase is in Pytorch. So given the time constraints, we could not reimplement the baselines ourselves and hence we use the reported numbers in the papers. We make comparisons to these methods by using the information provided in the papers and analyzing the official code. For each of these methods, we identify the protocol they used and compare their reported performance to SimpleView and PointNet++ in the closest protocol we evaluated. We can see that both SimpleView and PointNet++ perform as well as or better than prior methods, which corroborates  our conclusions.
>
> **“It might be good to combine the third contribution in the introduction with the second one?”**
>
> Thanks for the suggestion. We combine the third and second contributions in the revised version.

---

> > ### Author Response · Authors · 2020-11-22
> > **Authors' Response to Reviewer3 (Part 2/2)**
> >
> > **‘In the section of “our protocol”, this paper uses the fixed set of 1024 points instead of re-sampling different points at each epoch. Why?’**
> >
> > Either a fixed set or resampling is valid as long as it is the same choice for all methods compared. We chose a fixed set because it is the setting used by the very initial works (PointNet and PointNet++) that established the point cloud classification. In addition, a fixed set removes a source of randomness in our comparisons and potentially helps reduce the variance of results.
> >
> > **‘In the section of “generating depth images from point cloud”, this paper uses two strategies, one using the minimum depth of all points which makes sense to me. But the latter one which uses a weighted average of depth does not seem to be reasonable as it is essentially creating imaginary non-existing points which correspond to the averaged depth’**
> >
> > The strategy of using weighted average is a first order approximation for a more involved rendering strategy described in [1], whereby mean-shift clustering is used to estimate which points are on the surface. The depth of the points on the surface is averaged to get the depth at a pixel location. Though other strategies could be used, we find the weighted average strategy to be simple, efficient and effective (it works slightly better than the minimum depth strategy on the validation set). Note that our contribution lies in showing that such a simple and “dumb” baseline works surprisingly well and the design choices of the baseline can almost certainly be improved.  Exploring directions to extend SimpleView would be an exciting direction for future research.
> >
> > [1] Felix J¨aremo Lawin, Martin Danelljan, Patrik Tosteberg, Goutam Bhat, Fahad Shahbaz Khan, and Michael Felsberg. Deep projective 3d semantic segmentation. CAIP (2017).
> >
> > We would very much appreciate it if you could let us know if any concerns still remain. Thanks again for your suggestions and help in improving our submission.

---

### Official Review · AnonReviewer2 · 2020-10-31
**This paper revisits various training protocols and experimental settings for point cloud classification. Moreover, this paper proposes a simple yet effective projection-based SimpleView method, which achieves state-of-the-art performance on the ModelNet40 classification benchmark. However, I will consider it as a valuable technical report, instead of an appropriate academic paper for ICLR.**

**Rating:** 4
**Confidence:** 5

**Review:**

This paper discusses several protocols including data augmentation, point distribution, loss function, ensemble scheme, and testing models, which serves as a kindly reminder that the training protocol matters. As claimed, earlier work like PointNet++ can still achieve comparable performance to more recent methods. Such observations are useful, as different methods are supposed to be developed and measured under a unified setting. Moreover, the authors investigated a new projection-based SimpleView method by converting point clouds into depth images, achieving SOTA performance without pretrained CNNs.

Although such detailed reviews over the training protocols are helpful, the authors failed to propose any novel methods or evaluation metrics (notice the SimpleView is also a commonly used methods in other multi-view framework), resulting in relatively inadequate novelty and originality. In practice, when we evaluate or re-implement these deep point cloud networks, we naturally attempt to add and remove some basic tricks. I believe there are a lot of people already knowing what has been explored in this paper. Nevertheless, this work still has a great practical value that helps us quantitatively understand how different protocols influence several commonly-used models. Moreover, different methods are compared only for point cloud classification task on ModelNet40, which only rely on global feature and cannot fully demonstrate the representation ability among different methods. For SimpleView, providing the image resolution is preferred.  Additionally, some typos, e.g. in "randon" in Table 1, should also be corrected.

Pros:
This paper investigates protocols for various point cloud networks for point cloud classification, with an observation that simple PointNet++ and SimpleView can already achieve competitive performance

Cons:
This paper lacks academic insights and novelty, making it more like a technical report. Besides, it only focuses on the classification task, which cannot convincingly conclude the representation ability of different methods. Some typos and missing of experiment settings

---

> ### Author Response · Authors · 2020-11-18
> **Authors' Response to Reviewer2 (Part 1/2)**
>
> Thank you for your feedback and suggestions. It is encouraging that you found our work valuable. Following we address your concerns:
>
> **“Although such detailed reviews over the training protocols are helpful, the authors failed to propose any novel methods or evaluation metrics”;  “This paper lacks academic insights and novelty, making it more like a technical report.”;  “I will consider it as a valuable technical report, instead of an appropriate academic paper for ICLR.”**
>
> We respectfully disagree that our paper lacks academic insights and novelty. Our paper presents two surprising findings that were not present in the current literature  and contradict conventional wisdom.
>
> First, we show that auxiliary factors independent of network architecture make a huge difference in performance and obscure the effect of novel architecture. This goes against the perception that improvement in performance is primarily due to novel architectures. Second, we show that a very simple projection based method outperforms sophisticated point based models, which challenges the assumption in the field that point based methods are superior choices for point cloud classification.
>
> We respectfully disagree that the lack of a novel method or evaluation metric is necessarily a reason for rejection. There are different kinds of research contributions warranting publication. Proposing a novel method or metric is only one of them. A novel method or metric has never been a requirement for acceptance to ML conferences, in particular ICLR.
>
> In fact, papers similar to ours, which do not  propose any novel method or metric, but uncover important results by revisiting prior work, have been accepted to ICLR and other top-tier ML venues. For example, Melis et al. [1], in their ICLR 2017 paper, reevaluate neural methods in NLP and show how LSTMs outperform more recent methods.  Similarly, Dhillon et al. [2], in their ICLR 2019 paper, show how a simple baseline that remained mostly unnoticed outperforms many sophisticated few-shot algorithms. Similar papers which revisit the past progress and/or propose simple and effective baselines have also been published at other top-tier ML venues [3, 4].
>
> [1] Melis, Gábor, Chris Dyer, and Phil Blunsom. "On the state of the art of evaluation in neural language models." ICLR (2017).
>
> [2] Dhillon, Guneet S., Pratik Chaudhari, Avinash Ravichandran, and Stefano Soatto. "A baseline for few-shot image classification." ICLR (2019).
>
> [3] Martinez, Julieta, Rayat Hossain, Javier Romero, and James J. Little. "A simple yet effective baseline for 3d human pose estimation." ICCV (2017).
>
> [4] Musgrave, Kevin, Serge Belongie, and Ser-Nam Lim. "A metric learning reality check." ECCV (2020).
>
> **“In practice, when we evaluate or re-implement these deep point cloud networks, we naturally attempt to add and remove some basic tricks. I believe there are a lot of people already knowing what has been explored in this paper.”**
>
> We respectfully disagree that our results are widely known in the community. If the auxiliary factors we identified are widely known to significantly improve performance, we should expect them to be adopted or controlled for in most papers, especially the more recent ones, but evidence from existing papers suggests the opposite. For example:
>
> - One of our findings is that using the smooth loss can significantly improve performance. However, among the approaches we considered, only DGCNN used this smooth loss, and none of the earlier works (PointNet, PointNet++, PointCNN) and the later ones (RSCNN, DensePoint, Point-ASNL) used it. No paper mentioned it, including the DGCNN paper itself. We only found out through the open source code of DGCNN.
>
> - One of our findings is that removing the rotation augmentation used by earlier works (PointNet, PointNet++) significantly improves performance. Many recent methods (DGCNN, RSCNN, DensePoint) do not use rotation augmentation, but do not control for it when making comparisons with earlier works.
>
> - One of our findings is that in many recent methods (DGCNN, RSCNN, DensePoint), the model is evaluated on the test set after every epoch during training and the model with the best test performance is used for evaluation. However, in many earlier methods (PointNet, PointNet++) only the model after the last training epoch is used for evaluation. We find that this difference in the evaluation scheme makes a significant difference in performance, but we do not find any mention of controlling for this factor in the published papers.
>
> Finally, even if our results were widely known to experts, the fact that they are not apparent from existing literature would indicate a serious issue, making it even more valuable to have our paper published. We believe it would be troubling  and counterproductive if  important facts were  widely known to specialists but could not  be found by non-specialists through reading published papers.

---

> > ### Author Response · Authors · 2020-11-18
> > **Authors' Response to Reviewer2 (Part 2/2)**
> >
> > **“different methods are compared only for point cloud classification task on ModelNet40, which only rely on global feature and cannot fully demonstrate the representation ability among different methods.”**
> >
> > We do not compare only on ModelNet40. We also conduct experiments on ScanObjectNN, a real world point cloud classification dataset. We show how SimpleView outperforms state-of-the-art models on ScanObjectNN, as well as demonstrate how it achieves better cross-dataset generalization.
> >
> > Also, it is not the case that that point cloud classification task on ModelNet40 relies only on global features. When we reduce the resolution of image in SimpleView from (128 X 128) to (64 X 64), the performance decreases from 93.0 to 92.1. This shows that finer details of the object matter.
> >
> > **Image resolution for SimpleView is not provided.**
> >
> > Thanks for pointing it out. The image resolution for SimpleView is (128 X 128). We will add this information in a future version.
> >
> > **“Besides, it only focuses on the classification task, which cannot convincingly conclude the representation ability of different methods.”**
> >
> > We only consider the representation ability of different methods for point cloud classification. We do not make any claims about the representation ability of models for other tasks. Point cloud classification is a widely studied problem with real-world impact as it forms the core of object detection and retrieval systems. Exploring other point cloud processing tasks is out of scope of our work and an exciting direction for future work.
> >
> > **"some typos, e.g. in "randon" in Table 1, should also be corrected"**
> >
> > Thanks for pointing it out. We will revise in a future version. We will also carefully review and fix any other typo.
> >
> > **“missing of experiment settings”**
> >
> > We would very much appreciate it if you could let us know the specific experiment settings we are missing.
> >
> > We would very much appreciate it if you could  let us know if any concerns still remain. Thanks again for your suggestions and help in improving our submission.

---

### Comment · ~Francesco_Landolfi1 · 2021-03-14
**Please, revise your decision**

**Disclaimer:** I do not know any of the authors.

Although I understand that this is only an "application paper", this work shows that (alas, once again!), there is a serious reproducibility issue in the ML research. The authors did the "dirty job" of running all the experiments all over again in a fair setting, without making model selection on the test set, showing that many of current "SOTA"s in point cloud processing do not really deserve this definition.

I find very unjust the decision of rejecting this article solely on the fact that it has no academic novelty, since

 1. as shown by this very paper, many works that were considered "novelty" perform at most on par with simple baselines such as multi-view camera;
 2. this paper wouldn't have been written if researchers had followed the good ol' academic principles, such as not selecting the model that best performed on the test set.

The AC argued that this article highlights the problem without giving a solution. This is also untrue, since the author propose multiple protocols for a fair experimental setting, providing the possible combinations of models, data augmentations and loss functions, that researchers in the future can adopt to obtain a *comparable* result with the current, revised and *retested* "SOTA". They also *suggest*  which protocol is the best one and should be followed, contrarily to what is stated in the AC comment.

R2 suggests that this could be better suited as a technical report. I agree, in the sense that this could be a good technical report, but this is not a good point for rejecting this article, in my opinion. First, there are other examples of articles like this, even one published in this very conference: in [1], the authors showed that a simple MLP baseline over-perform most GNN models. Second, providing a new "ranking" of the current models is of great importance to novel research, since new valid works in the same field might not be published since they do not stand against unfair results.

Lastly, but not by importance, I find very unjust the decision of the AC to reject this paper after 3/4 of the reviewers voted for an accept, with 7-7-7-4. Although it might be the right thing to do in some circumstances, going against the decision of a committee should be accompanied by a solid motivation. The only point of the AC in their comment is that the experiments alone are not enough for carry the paper, even though the authors empirically prove their point that a simple baseline beats the current state-of-the-art.

[1] F. Errica, M. Podda, D. Bacciu, A. Micheli. *"A Fair Comparison of Graph Neural Networks for Graph Classification"*. ICLR 2020.

---

### Decision · Program_Chairs · 2021-01-07
**Final Decision**

**Decision:**

Reject

**Comment:**

This paper received three recommendations of accept and one recommendation of reject.   The paper is mixed.  The results presented are both compelling and will have impact on the community.  The AC does not agree with R2's views that the paper requires proposal of a novel method for acceptance.  At the same time, the AC also does not agree with the views of the other reviewers that the current experiments alone are enough to carry the paper without more conclusive statements.  As hinted by R3, simply pointing out the problems is not enough without proposing how to adjust our models and experimentation protocols in the future is insufficient.

In its current state, the paper would make for a good workshop submission.  Alternatively, the AC suggests to the authors to expand on the SimpleView baseline and or propose alternative solutions or protocols.